# Pd Nanoparticles Loaded on Cu Nanoplate Sensor for Ultrasensitive Detection of Dopamine

**DOI:** 10.3390/s24175702

**Published:** 2024-09-02

**Authors:** Haihu Tan, Xuan Zhang, Jinpu Xie, Zengmin Tang, Sijia Tang, Lijian Xu, Pingping Yang

**Affiliations:** 1College of Packaging and Materials Engineering, Hunan University of Technology, Zhuzhou 412007, China; tanhaihu2020@hut.edu.cn (H.T.); 18856765173@163.com (J.X.); 2College of Life Science and Chemistry, Hunan University of Technology, Zhuzhou 412007, China

**Keywords:** Pd nanoparticles, two–dimensional Cu nanoplates, dopamine, electrochemical active surface areas, sensor

## Abstract

The detection of dopamine is of great significance for human health. Herein, Pd nanoparticles were loaded on Cu nanoplates (Pd/Cu NPTs) by a novel liquid phase reduction method. A novel dopamine (DA) electrochemical sensor based on the Pd NPs/Cu/glass carbon electrode (Pd/Cu NPTs/GCE) was constructed. This sensor showed a wide linear range of 0.047 mM to 1.122 mM and a low limit of detection (LOD) of 0.1045 μM (S/N = 3) for DA. The improved performance of this sensor is attributed to the obtained tiny Pd nanoparticles which increase the catalytic active sites and electrochemical active surface areas (ECSAs). Moreover, the larger surface area of two–dimensional Cu nanoplates can load more Pd nanoparticles, which is another reason to improve performance. The Pd/Cu NPTs/GCE sensor also showed a good reproducibility, stability, and excellent anti–interference ability.

## 1. Introduction

As an important neurotransmitter, dopamine (DA) plays a crucial role in the mammalian central nervous system [1,2,3,4,5,6,7,8]. The imbalance of DA metabolism in the body causes it to be prone to many diseases, such as Parkinson’s disease, schizophrenia, epilepsy, etc. Developing a rapid, accurate, and effective method for the detection of DA is of great significance for physical health monitoring and clinical diagnosis. Currently, various analytical methods have been established for DA detection, including capillary electrophoresis, chemiluminescence, fluorescence, chromatography, and electrochemical methods [9,10,11,12]. Among these methods, the electrochemical method has been attracting concentrated attention due to its considerable advantages, such as high sensitivity, good selectivity, relatively low instrument cost, and easy miniaturization to realize real-time detection [13]. DA molecules contain two phenolic hydroxyl groups, which are electrochemically active and can be oxidized easily on the surface of the electrode [14]. Recently, various electrochemical sensors have been designed for DA detection. The chemically modified glass carbon (GC) electrode, as one of the important components in electrochemical sensors, can recognize, collect, and transmit signals; its properties are greatly affected by the catalyst [15,16].

Nanostructured metals, including copper (Cu), noble metal, multi-metallic nanomaterials, three–dimensional hydrogel, etc. [17,18,19,20,21,22,23], have been widely employed for the fabrication of electrochemical sensors due to their prominent intrinsic conductivity and high catalytic activity. Among many metallic nanomaterials, Cu has been widely used in the construction of DA electrochemical sensors to improve its electrode performance, due to its low price, excellent conductivity, and catalytic activity [24,25]. Sükriye Ulubay et al. reported the application of nano–Cu/PPy/GCE for the electrochemical detection of DA in the experimental and real samples (human urine samples) [26]. The catalytic activity of the modified GC electrode for DA oxidation was improved by the incorporation of Cu nanoparticles on the electrode surface. Later, Cu nanoparticles have already received much attention and usually worked as catalytic active substances that collaborated with carbon nanomaterials, graphene, graphene oxide, reduced graphene oxide, etc. More recently, our group also modified the GC electrode with anisotropic Cu nanoplates for the detection of DA [27]. The oxidation peak current was linearly correlated with the DA concentration in the range of 0.2 mM to 2.21 mM with a detection limit (LOD) of 62.4 μM (S/N = 3). Moreover, the prepared Cu/GCE electrode showed a good anti-interference performance in the presence of hydrogen peroxide (H_2_O_2_), ascorbic acid (AA), and glucose (Glu) at a high concentration. However, due to the easy oxidation of Cu in the air and work environment, the physical and chemical properties of copper nanoplates still need to be improved. To address this issue, multi-metallic hybrid nanomaterials have recently been synthesized by loading the second metal, such as silver (Ag) nanoplates, Ag nanowires, and Cu nanowires coated with a noble metal, on the anisotropic metal nanoparticle (nanowire, nanoplates, etc.). Liu A. et al. constructed the gold (Au) shell on the surface of Ag nanoplates to obtain Ag@Au core-shell triangular nanoplates, which possessed a better stability on the colorimetric sensing of glucose than that of Ag triangular nanoplates [28]. Zhang B. et al. found that the Cu@Ag core–shell nanowires showed a high stability even at high reaction temperatures and high humidity, due to the protection of Ag on the surface of Cu nanowires [29]. In addition, the catalytic activity of multi-metallic hybrid nanomaterials can also be regulated due to the potential strain effect, generated interface, and synergistic effect. Li S. et al. decorated the surface of the Cu nanowire with a palladium (Pd) shell and confirmed the enhanced activity and stability of Cu@Pd nanowires for formic acid oxidation due to the synergetic effect of Cu and Pd [30]. Pd also shows a good catalytic activity in the electrochemical oxidation of DA [31,32,33]. Huang J. et al. prepared carbon nanofibers (CNFs) loaded with Pd nanoparticles by electrospinning and heat treatment processes. The constructed Pd/CNF–CPE has good electrochemical catalytic activity for DA, uric acid (UA), and AA [34]. Chellakannu Rajkumar et al. reported the facile preparation of highly stable Pd nanoparticles loaded on porous carbon aerogel (CA) via microwave reduction. The resulting Pd/CA–GC electrode showed a wide linear response range (0.01–100 µM) and low LOD (0.0026 µM) for sensing DA [35]. Ming-Hung Chiang et al. reported the palladium/copper popcorn nanoparticles as sensors for the differential pulse voltammetric determination of DA, where the Pd/Cu popcorn nanoparticles are the alloy structure [36]. However, to our best knowledge, Pd-nanoparticle-decorated Cu nanoplates’ metallic hybrid nanomaterials have rarely been used to construct DA electrochemical sensors.

Herein, we reported the facile preparation of Pd nanoparticles loaded on Cu nanoplates (Pd/Cu NPTs/GCE) and their application on the electrochemical detection of DA. The influences of the amount of polyvinyl pyrrolidone (PVP) and the Pd precursor on Pd dispersion on the surface of Cu nanoplates were discussed. The performance of the Pd/Cu NPTs/GCE electrode in detecting dopamine was evaluated by several electrochemical tests.

## 2. Experimental Method

### 2.1. Reagents

Disodium tetrachloropalladate (Na_2_PdCl_4_, 99%), cupric bromide (CuBr_2_, 99%), polyvinyl pyrrolidone (PVP, MW: 10,000), glucose (Glu, C_6_H_12_O_6_, 99%), ascorbic acid (AA, C_6_H_8_O_6_, 99%), dopamine hydrochloride (C_8_H_11_NO_2_·HCl, 98%), disodium hydrogen phosphate (Na_2_HPO_4_, ≥99%), Nafion solution (perfluorinated resin solution, 5%, wt%), sodium dihydrogen phosphate (NaH_2_PO_4_, 99%), hydrogen peroxide(H_2_O_2_, 30 wt%, wt%), uric acid (UA, 99%), and branched polyethyleneimine (BPEI, 50 wt%, MW: 60,000) were obtained from InnoChem (Beijing, China) and used without purification. Deionized water (DI water) was used throughout the experiments.

### 2.2. Synthesis of Pd/Cu NPTs Nanoplates

Firstly, the Cu nanoplates were synthesized by the seed-assisted hydrothermal method [27]. Typically, 0.1 mmol CuBr_2_ and 20 mg BPEI (Appendix A) were dissolved in 7 mL of deionized water to obtain a blue transparent solution in a 90 °C water bath. Then, 40 µL of pre-prepared Ag nanoparticle suspension with a concentration of 3 mM was added to the above blue solution, followed by the addition of 3 mL of ascorbic acid solution (0.6 M). Stirring of the mixed solution was maintained during the overall process. After 2 h, a brownish-red reaction suspension containing Cu nanoplates was obtained and cooled down to room temperature, and then the Cu nanoplates were collected and washed (Cu NPTs). The Cu NPTs were then redispersed into 5 mL of DI water containing AA and PVP. The AA can effectively prevent the oxidation of Cu during the storage and Na_2_PdCl_4_ reduction. Subsequently, Na_2_PdCl_4_ solution (10 mM) was added into the Cu nanoplates aqueous suspension and magnetically stirred for 10 min at ambient temperature. By reduction of Na_2_PdCl_4_, Pd nanocrystals were formed and deposited on the surface of Cu nanoplates. Simultaneously, the metallic luster of Cu nanoplates gradually faded. The volume of Na_2_PdCl_4_ solution was varied from 20 to 70 μL to adjust the load percentage of Pd on the surface Cu nanoplates. Finally, the Pd nanoparticles were loaded on the Cu nanoplates (Pd/Cu NPTs).

### 2.3. Electrochemical Detection of DA

Pd/Cu NPTs/GCE was fabricated by dropping 7 µL of Pd/Cu NPT suspension on the pretreated GCE and dried at ambient temperature. The pretreated GCE is prepared as follows: GC electrode was polished with 5.0, 1.0, and 0.3 μm Al_2_O_3_ slurry in turn and washed ultrasonically with water and ethanol for 5 min, respectively. Then, 5 µL of Nafion solution was dropped onto the surface of Pd/Cu NPTs/GCE and was allowed to evaporate at room temperature for 20 min. The electrochemical performance of Pd/Cu NPTs/GCE was evaluated by conventional electrochemical techniques including electrochemical impedance spectroscopy (EIS), cyclic voltammetry (CV), and chronoamperometry (CA). All the electrochemical measurements were carried out on a CHI 760e electrochemical workstation (Shanghai Chenhua Inc., Shanghai, China) using a classic three-electrode system, where Pd/Cu NPTs/GCE, platinum plate, and saturated calomel electrode were used as the working, counter, and reference electrodes, respectively. For electrochemical tests, the three–electrode assembly was immersed into a 10 mL electrochemical cell containing phosphate-buffered saline (PBS, 0.2 M, pH = 7.2) and the appropriate concentration of DA.

### 2.4. Material Characterization

The morphology and elemental distribution of Pd/Cu NPTs was observed by scanning electron microscopy (SEM, Zeiss Gemini 300, at 200 kV, Oberkochen, Germany) and transmission electron microscope (TEM, at 200 kV, JEOL JEM–2100, Tokyo, Japan) with energy-dispersive spectroscopy (TEM–EDS, at 200 kV, JEOL JEM–2100). The crystal structure of Pd/Cu NPT nanoplates was analyzed by X-ray diffractometer (XRD, Bruker D8, Karlsruhe, Germany).

## 3. Results and Discussion

### 3.1. Preparation and Characterization of Pd/Cu NPTs

In a typical preparation as shown in Figure 1, pre-prepared two-dimensional Cu nanoplates were used as support to fabricate Pd/Cu NPTs. In the Cu nanoplates aqueous suspension, the injected Na_2_PdCl_2_ reacted with AA in the presence of PVP and was reduced to Pd, which then adhered to the surface of Cu nanoplates and grew to Pd/Cu NPTs. Afterward, the Pd/Cu NPTs/GC electrode was fabricated through a facile drop-casting method and was employed as a working electrode for DA detection.

As shown in Figure 2a and Appendix A, the Pd/Cu NPTs were synthesized in an aqueous solution and the Cu nanoplates were synthesized through the seed-assisted hydrothermal method reported in the previous literature [27]. The UV–Vis absorption spectra of the Cu nanoplate suspension (the black curve in Appendix A) and Pd/Cu NPT suspension (the red curve in Appendix A) did not present any UV absorption peak due to the too large size of Cu. Therefore, there is no corresponding Cu absorption peak in either material. The synthesized Cu nanoplates have a smooth surface (Figure 2b and Appendix A) which presents a pink color with a metallic luster. The 20 μL Na_2_PdCl_4_ solution (10 mM) was injected into the suspension of Cu nanoplates in the presence of AA and PVP; the Cu nanoplates were evenly coated with Pd nanoparticles. The thickness of Cu nanoplates was around 270 nm (Appendix A). The corresponding Cu size distribution is obtained by calculating 200 Cu nanoplates (Appendix A), which is around 14.6 ± 1.0 μm. After coating the Pd nanocrystals, the color of the suspension changes from pink to dark (Figure 2a), while the morphology of nanoplates does not change (Figure 2c). As shown in Figure 2d and Appendix A, many small nanoparticles with less than 10 nm are evenly distributed on the surface of the nanoplates with a uniform size distribution, confirming Pd nanoparticles were successfully loaded onto the surface of Cu nanoplates.

TEM was used to further investigate the microstructure of Pd/Cu NPTs. Figure 3a displays many small dark spots coated on the surface of nanoplates and Figure 3b reveals that the 2.20 Å lattice distance corresponds to the facet (111) of the Pd nanocrystals. The TEM-EDS analysis was employed to characterize the distribution of red spots with a high density in Figure 3c, indicating the presence of Cu, and the scattered distribution of the green spots in Figure 3d indicates the presence of a relatively low concentration of Pd loading. The size distribution of Pd nanoparticles on Cu nanoplates was calculated and the average size is about 3.6 nm (Appendix A). Moreover, the 2.09 Å lattice distance corresponds to the facet (111) of the Cu nanocrystals (Appendix A). The EDS profile in Figure 3e shows the atomic ratio of 98.15:1.85 for Cu: Pd. In addition, the XRD was used to further confirm the crystal structure of the samples. As shown in Figure 3f, three diffraction peaks were observed at 43.30°, 50.35°, and 74.10°, which are assigned to the (111), (200), and (220) facets of Cu (JCPDS # 04–0836). The only crystalline Cu phase is identified due to the low atomic ratio of Pd in the Pd/Cu NPTs.

The dispersion of Pd nanoparticles on the surface of Cu nanoplates is highly dependent on the amount of the PVP and Na_2_PdCl_4_ precursor solution. In the absence of PVP, aggregated Pd nanoparticles shown in the inset of Figure 4a also were found at the side of the nanoplates (at the location marked by the yellow circle in Figure 4a). In addition, the SEM image in Figure 4b also reveals that some nanoparticles unevenly distribute on the surface of the nanoplates and have a big size. As a nonionic polymer, PVP not only can effectively avoid the agglomeration of Pd nanoparticles but also can significantly improve the size distribution and dispersion of nanoparticles on the surface of Cu nanoplates. Furthermore, as the additive amount of the Na_2_PdCl_4_ solution increased from 20 to 70 μL, Pd nanoparticles are aggregated and agglomerated due to the stacking of excessive Pd nanoparticles (Figure 4c,d, Appendix A). Nevertheless, the excessive and aggregated Pd nanoparticles cannot reduce the use amount of the noble metal, and may also affect the catalytic performance of Pd/Cu NPTs even at a higher Pd loading.

### 3.2. Electrochemical Sensing Properties of DA on Pd/Cu NPTs/GCE Electrode

The drop-casting method was applied to construct the Pd/Cu NPTs/GCE electrode with the assistance of the Nafion film. To initially verify the enhanced electrocatalytic activity of Pd/Cu NPTs/GCE, the electrochemical response of the Cu/GCE and Pd/Cu NPTs/GCE electrodes to DA were firstly analyzed using the DPV method in PBS solution (pH = 7.2). Moreover, we tested the dependence of the dopamine oxidation current on Pd/Cu NPTs/GCE at different pH (6.0, 6.4, 6.8, 7.2, and 7.6) in 0.2 M PBS containing 5.0 mM DA; we can see that pH = 7.2 is the most appropriate (Appendix A). As shown in Figure 5a, the Cu/GCE electrode possesses one characteristic oxidation peak of Cu at a potential of −0.1 V in the absence of DA. As the test is conducted in the presence of DA (5 mM), the Cu/GCE electrode shows a new characteristic oxidation peak to DA at a potential of around 0.2 V, indicating Cu nanoplates have a catalytic effect on the oxidation of dopamine and the Cu/GCE electrode also can recognize the presence of DA. Furthermore, the current response to DA is significantly enhanced using the Pd/Cu NPTs/GCE electrode instead of the Cu/GC electrode. Obviously, the addition of Pd improves the response performance of the sensor to DA due to the newly added active sites centered on Pd on the surface of the Cu nanoplates. Moreover, we provide the DPV–response of bare GCE in the presence/absence of DA (Appendix A). The impedance curves in Figure 5b show that the Pd/Cu NPTs/GCE and Cu/GCE electrodes have little difference in internal resistance. The electrode resistances of Cu/GCE and Pd/Cu NPTs/GCE were calculated to be 49.6 and 48.6 Ω, respectively. However, the former possesses a bigger slope compared with the latter due to the presence of Pd. Secondly, the influence of the volume of the Na_2_PdCl_4_ precursor solution on the DA electrochemical sensing properties was further investigated to verify the enhancement of Pd on the electrocatalytic activity of Pd/Cu NPTs. The Pd/Cu NPTs synthesized with 5, 20, 30, and 70 μL of Na_2_PdCl_4_ are marked as Pd/Cu NPTs–5, Pd/Cu NPTs–20, Pd/Cu NPTs–30, and Pd/Cu NPTs–70, respectively. Figure 5c,d show the DPV responses and the corresponding anodic peak current of 5 mM DA on different Pd/Cu NPTs/GCE electrodes. As shown in Figure 5c, a characteristic anodic peak appears at 0.1–0.3 V, which reveals the catalytic activity of Cu in the electrochemical oxidation of DA [27]. The Pd/Cu NPTs/GC electrode shows a noticeable increase in the anodic peak current of DA (Figure 5d). The maximum anodic peak current of DA reached the Pd/Cu NPTs–20/GC electrode, indicating Pd/Cu NPTs–20 showed a superior catalytic activity toward DA oxidation. The excellent electrocatalytic activity may be attributed to the uniform distribution of the obtained tiny Pd nanoparticles on the surface of the copper nanosheets, forming numerous island structures. The synergistic catalysis performances arise from the combined effects of catalytic superposition, interface interactions, and surface electron effects [36]. Loading the appropriate amount of Pd metal on the surface of the Cu nameplates not only can preserve the catalytic activity of the copper nanoplates but also can increase the catalytic active center with Pd, thus achieving an enhanced catalytic activity of the Pd/Cu NPTs for the electro-catalytic oxidation of dopamine. Compared with the aggregation of Pd in Pd/Cu NPTs–70, Pd/Cu NPTs–20 may be capable of providing more effective active sites. However, the anodic peak current of DA decreases gradually with further increasing the volume of the Na_2_PdCl_4_ precursor solution, probably due to the excessive loading of aggregated Pd nanoparticles, which seriously weakens the synergistic effect between Cu and Pd [36]. Therefore, the 20 μL Na_2_PdCl_4_ precursor solution was used for the synthesis of Pd/Cu NPTs in the subsequent experiments. We also performed a TEM analysis of Pd/Cu NPTs–20 (Appendix A). The higher catalytic activity is attributed to the loading of more Pd nanoparticles on Cu nanoplates. Moreover, the electrochemically active surface area (ECSA) measured with bare GCE, Cu/GCE, and Pd/Cu NPTs/GCE are 0.0702, 0.071, and 0.0912 cm^2^, respectively (Appendix A). The CV curves of 5 mM DA were recorded on the Pd/Cu NPTs/GC electrode at different scan rates (Figure 5e). As the scanning rates increase from 10 to 60 mV/s, the anodic peak current of DA linearly increase with the scanning rates (Figure 5f,g). The corresponding linear equation is expressed as I_ox_ (μA) = 3.35v (mV/s) + 10.69 with a correlation coefficient (R^2^) of 0.9993, which suggests that the electrochemical oxidation of DA on the surface of the Pd/Cu NPTs/GC electrode is dominated by surface adsorption. Moreover, the Voltage value (P_ox_) of the oxidation peak also shifts and shows a linear relationship to the Napierian logarithm of the scan rate (ln*v*), expressed as P_ox_ = 0.024ln*v* + 0.309 (R^2^ = 0.9812). Based on the equation [37] of P = P° + (RT/anF)ln(RTk°/anF) + (RT/anF)ln*v* (where T is for the Kelvin temperature, R is for the ideal gas constant, k° is for the heterogeneous electron transfer rate, F is for Faraday constant, P° is for the formal potential, a is 0.5 for an irreversible process, and n is for transferred number of electrons), the n value can be figured out and is around 2, which indicates that two electrons are included in the process of the electrochemical oxidation of DA [38]. Furthermore, the DPV responses of various concentrations of dopamine were recorded on the Pd/Cu NPT electrode, and the results are shown in Figure 5h. The response peak current of DA gradually increases with the increase in DA concentration. In addition, there is a good linear relationship between the anodic peak current and DA concentration in the range of 0.047–1.122 mM (Figure 5i). The corresponding linear regression equation is I(µA) = 15.29C_DA_ (mM) + 12.92 (R^2^ = 0.9909). The LOD is calculated to be 0.1045 μM. Compared with previously reported electrodes (Table 1), the developed Pd/Cu NPTs/GCE shows superior performance with a lower detection limit of 0.1045 μM in the concentration range of 0.047–1.12 mM.

### 3.3. Anti-Interference Ability of Pd/Cu NPTs/GC Electrode

An important aspect to be considered in the manufacture of DA electrochemical sensors for the early diagnosis of neurological diseases is to effectively avoid the redox of potential interfering substances. The response current of 1.0 mM DA in the presence of AA (1.0 mM) and UA (1.0 mM) are shown in Figure 6, Appendix A. At a working potential of 0.2 V, DA is first added to check the current response, and then different interfering substances were continuously added to the stirred PBS solution. The steady-state current is reached within 3 s, suggesting a fast response of the proposed sensor. AA, UA, and Glu often coexisted with DA in physiological samples such as human serum. Therefore, these substances are selected as the potential interfering substances for the anti-jamming investigation. To our surprise, there is no significant change in the response current of DA when coexisting with these interfering species, indicating that the proposed dopamine sensor has an excellent anti-interference ability.

### 3.4. Repeatability and Stability of Pd/Cu NPT Electrode

As shown in Figure 7a, the CV curves of 1.0 mM DA recorded at four independent Pd/Cu NPT electrodes almost overlap. The corresponding response peak current of DA on these four electrodes is shown in Figure 7b. The relative standard deviation (RSD) is 3.31%, indicating that the fabrication of Pd/Cu NPT electrodes is highly reproducible. It is well-known that good stability is essential for practical applications. As shown in Figure 7c,d, the response current of DA decreases by 3.3% over 50 multi–CV cycles, indicating the much better stability of Pd/Cu NPTs than that of Cu/GCE [27]. Compared with copper, palladium (Pd) has better stability. In the island structure composite that is formed, the stable Pd layer partially covers the interior of Cu, creating a barrier that isolates oxidizing active substances. This ultimately improves the stability of the Pd/Cu NPTs in the electrochemical sensing of DA [44].

### 3.5. Detection in Human Serum Sample

Since Bn often has been used as a medicament, to evaluate the applicability of the manufactured electrochemical sensor in actual samples, the Pd/Cu NPT sensor has detected the DA in human serum through a standard addition method. Human serum samples have been obtained from the Zhuzhou City Second Hospital and diluted 100 times with PBS (pH 7.2). The Pd/Cu NPT sensor has obtained the oxidation current in the blank sample and the spiked sample, and the DA concentration is calculated through the calibration equation. Appendix A shows that the recovery rate of Bn is 97.5–100.0%, which indicates that Pd/Cu NPTs can be used to the detection of Bn in human serum.

## 4. Conclusions

In summary, Pd nanocrystals were uniformly loaded on the surface of Cu nanoplates by controlling the amount of the PVP and Na_2_PdCl_4_ precursor. The electrochemical measurements demonstrated the suitable loading amount of Pd effectively enhanced the electrochemical sensing performance of DA and avoided the oxidation of Cu nanoplates. The Pd/Cu NPT electrode showed a higher response peak current than Cu/GCE, mainly due to the synergistic effect between Cu and Pd. As a result, the Pd/Cu NPT electrode showed a wide linear response ranging from 0.047 to 1.122 mM with a low LOD of 0.1045 μM (S/N = 3). Moreover, the Pd/Cu NPT electrode showed an excellent anti-interference ability, reproducibility, and long-term stability.

## Figures and Tables

**Figure 1 sensors-24-05702-f001:**
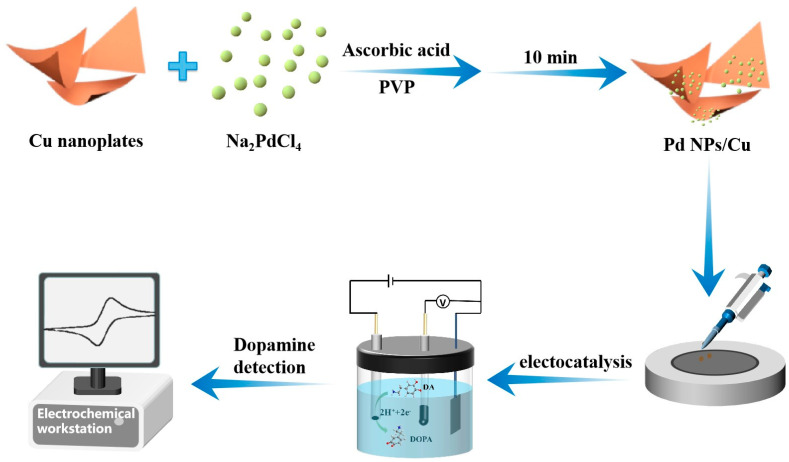
The schematic diagram for the synthesis of Pd/Cu NPTs and electrochemical detection of DA.

**Figure 2 sensors-24-05702-f002:**
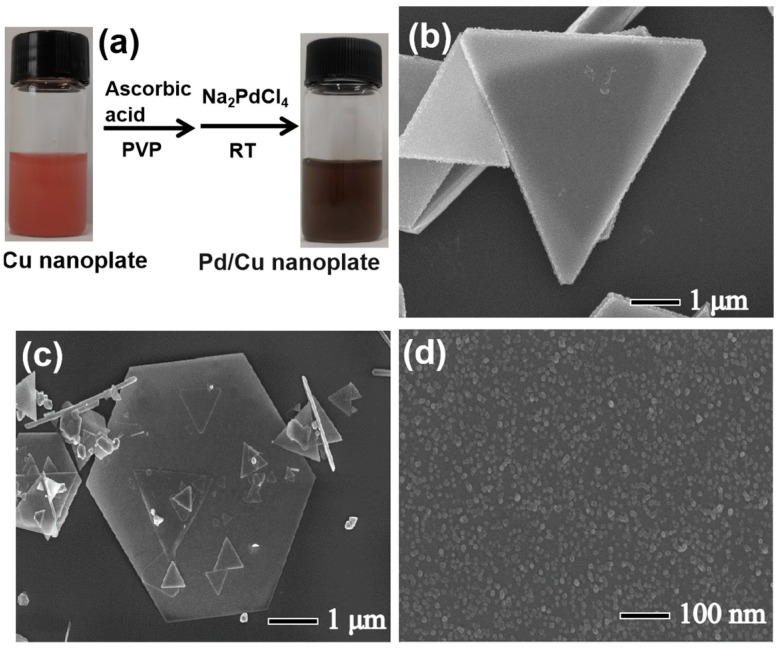
(**a**) Schematic illustration of the synthesis of Pd/Cu NPTs; SEM image of (**b**) Cu nanoplates and (**c**) Pd/Cu NPTs; and (**d**) high–resolution SEM image of Pd/Cu NPTs.

**Figure 3 sensors-24-05702-f003:**
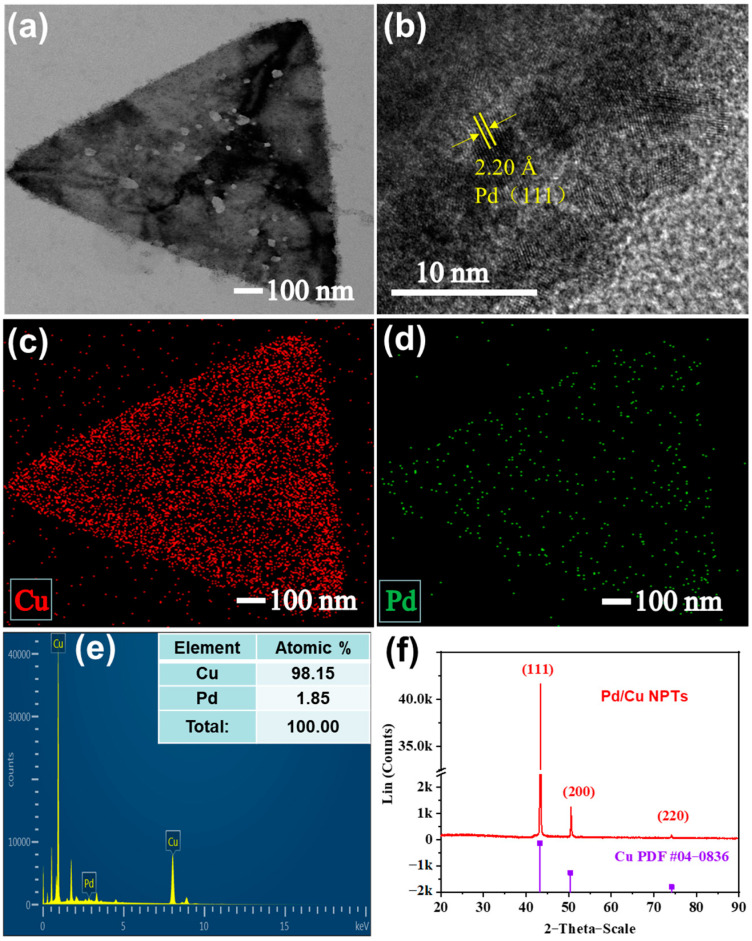
(**a**) TEM image and (**b**) high–resolution TEM image of the Pd/Cu NPTs; TEM-EDS mapping images of (**c**) Cu and (**d**) Pd, and (**e**) TEM–EDS profile and (**f**) XRD patterns of Pd/Cu NPTs.

**Figure 4 sensors-24-05702-f004:**
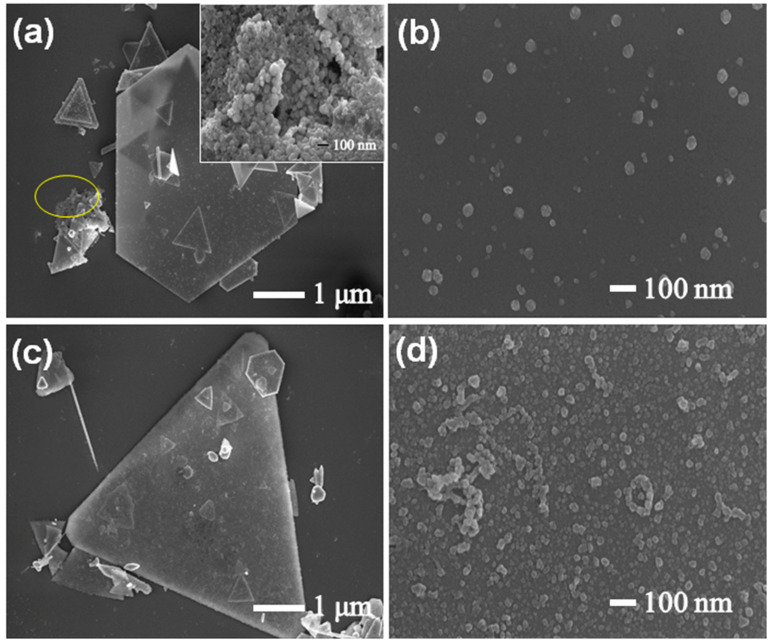
(**a**,**b**) SEM images of nanoplates synthesized in the absence of PVP, and (**c**,**d**) SEM images of Pd/Cu NPTs obtained as the additive amount of Na_2_PdCl_4_ solution is 70 μL.

**Figure 5 sensors-24-05702-f005:**
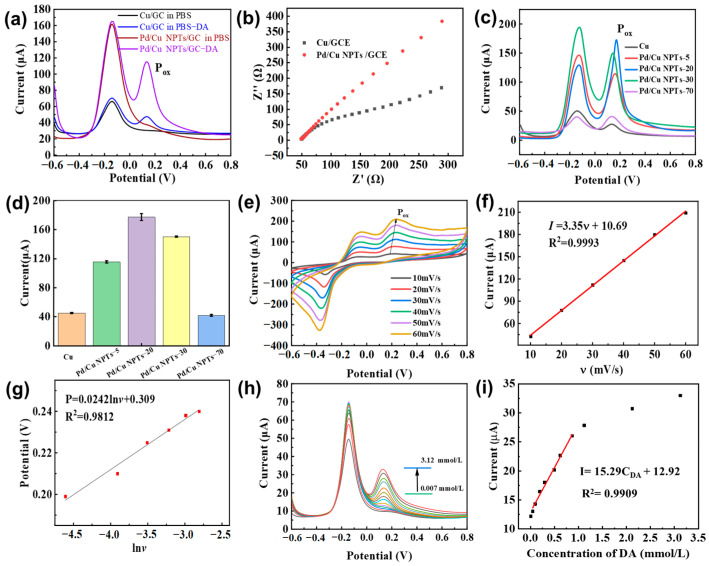
(**a**) DPV response of Cu/GC and Pd/Cu NPTs/GC electrode in the presence/absence of DA; (**b**) impedance curves of the Pd/Cu NPTs/GC and Cu/GC electrodes; (**c**) DPV response and (**d**) the corresponding peak current of 5 mM DA on different Pd/Cu NPTs/GC electrodes; (**e**) CV curves of 5 mM DA on the Pd/Cu NPTs/GC electrode recorded at different scanning rates; (**f**) linear plot of the anodic peak current of DA versus scanning rates; (**g**) linear plot of the anodic peak potential of DA versus Napierian logarithm of scan rate; (**h**) DPV responses of various concentration DA recorded on the Pd/Cu NPTs/GC electrode; and (**i**) calibration curve of the anodic peak current of DA versus DA concentration.

**Figure 6 sensors-24-05702-f006:**
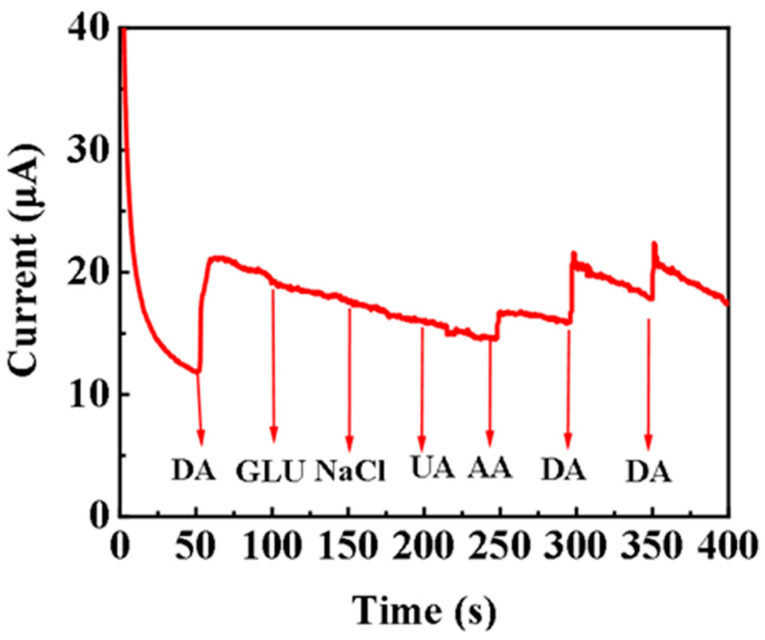
Response current signal of DA in the presence of potentially interfering substances.

**Figure 7 sensors-24-05702-f007:**
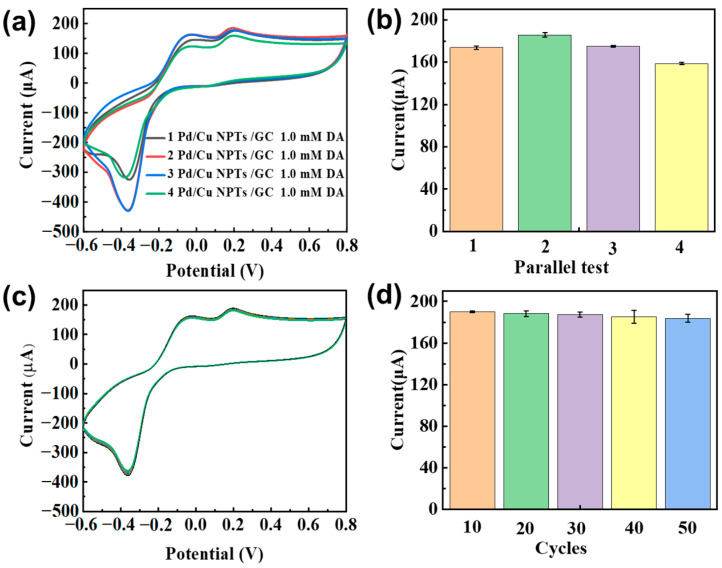
(**a**) CV curves of 1.0 mM DA recorded on four independent Pd/Cu NPT electrodes; (**b**) the corresponding bar graphs of response current of DA; (**c**) multi–CV scanning of 1.0 mM DA recorded on the Pd/Cu NPT electrode; and (**d**) the corresponding bar graphs current of DA in the multi–CV scanning.

**Table 1 sensors-24-05702-t001:** Comparison of the sensing performance of DA with previously reported electrodes.

Electrodes	Method	Linear Detection Range	LOD	Refs
Graphene/GCE	DPV	4–100 μM	2.64 μM	[39]
Fe_3_O_4_@Au–Gr/GCE	DPV	0.5–50 μM	0.65 μM	[40]
N–Cu–MOF/GCE	DPV	0.0005–1.78 mM	0.15 nM	[41]
Pt/CeO_2_@Cu_2_O	CV	0.5–100 μM	0.079 μM	[42]
Au–Cu_2_O/rGO/GCE	DPV	10–90 μM	3.90 μM	[43]
Pd/Cu NPTs/GCE	DPV	0.047–1.12 mM	0.1045 μM	This work

## Data Availability

Data are contained within the article.

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
