# Peer review of "Pd Nanoparticles Loaded on Cu Nanoplate Sensor for Ultrasensitive Detection of Dopamine"

_sensors, 2024, doi:10.3390/s24175702_

Round 1

Reviewer 1 Report

Comments and Suggestions for Authors

In this paper, the authors synthesized Pd nanocrystals-loaded Cu nanoplates, which was further utilized for the electrochemical detection of dopamine,

Generally, this kind of paper has been published for over ten years, with no novelty at alll, such as Electroanalysis 201729, 2106., https://doi.org/10.1016/j.jcis.2022.09.139, etc. And the authors are still carrying out similar research without solving obvious scientific problems. Why the plate could improve the detection efficiency? The reviewer find very little contribution of this paper to the field. So the reviewer can not recommend the publication of this work. 

Comments on the Quality of English Language

Could be improved.

Author Response

Comments 1In this paper, the authors synthesized Pd nanocrystals-loaded Cu nanoplates, which was further utilized for the electrochemical detection of dopamine,

Generally, this kind of paper has been published for over ten years, with no novelty at all, such as Electroanalysis 2017, 29, 2106., https://doi.org/10.1016/j.jcis.2022.09.139, etc. And the authors are still carrying out similar research without solving obvious scientific problems. Why the plate could improve the detection efficiency? The reviewer find very little contribution of this paper to the field. So the reviewer can not recommend the publication of this work.

Response1: We thank the reviewer for the kind advice. We have carefully thought about this statement.

This work focuses on the preparation of a new sensor. Firstly, we use a new type of support (Cu nanoplates) for Pd nanoparticles, which is also prepared by a relatively simple and environmentally friendly method, refer to these literatures (ChemistrySelect, 2(2017), 4655-4661; Life, 2022, 12, 999). By improving the method described in this paper, cu nanoplates can be synthesized within one hour and are easy to collect. Secondly, there are more materials with two-dimensional structure, and most of the current articles use two-dimensional carbon materials (graphene, graphene oxide, etc.) as a carrier to load precious metals to explore the synergistic effect between precious metals and carbon. Based on this, this work intends to use Cu nanoplates with two-dimensional structure as a substitute for two-dimensional carbon materials, and finally load Pd nanoparticles to obtain Pd/Cu NPTs composite materials. The catalytic activity of Pd/Cu NPTs composites was investigated by means of sensors. In view of the fact that Cu has good catalytic activity in other sensing fields, we continue to promote the relevant later work to explore the electrochemical catalytic performance of Cu nanoplates combined with other precious/rare metals, and do some technical and theoretical accumulation

In addition, we would like to thank you for your valuable comments, and in the next research work, we will continue to consider the innovation of the research. Thank you again for your suggestion of our work, and we will try to do better.

Reviewer 2 Report

Comments and Suggestions for Authors

This paper reports the development of a dopamine electrochemical sensor based on Pd NPs/Cu NPTs/glassy carbon electrode.

Questions and comments:

1. The Introduction should discuss recent advances in electrochemical detection of dopamine. For example, see the following reviews: Biosensors, 2021, 11(6), 179, https://doi.org/10.3390/bios11060179; Trends in Analytical Chemistry, 2023, 169, 117367, https://doi.org/10.1016/j.trac.2023.117367; Sensors and Diagnostics, 2023, 2, 559-581, https://doi.org/10.1039/D2SD00230B; Journal of Electroanalytical Chemistry, 2024, 959, 118157, https://doi.org/10.1016/j.jelechem.2024.118157; ECS Sensors Plus, 2024, 3(2), 020601, https://doi.org/10.1149/2754-2726/ad3950.

2. Throughout the entire manuscript, it is necessary to adhere to the unambiguous designation of the glassy carbon electrode (GCE) and the sensor developed on its basis. Note that line 182 uses "Cu@Pd" while Table 1 uses "Pd@Cu/GC".

3. Lines 46-47: note that Ulubay et al. used "nano-Cu/PPy/GCE" and "human urine samples" instead of "Cu/GC" and "human blood serum".

4. Line 74: note that Huang et al. used "Pd/CNF-CPE" instead of "Pd/CNFs modified GC".

5. The abbreviations PVP (line 89), PBS (line 125) and Glu (line 254) should be identified. Subsection 2.1. "Reagents" does not contain information on "Glu".

6. Line 93: "uric acid (C5H4N4O3, 99%)" should be replaced with "uric acid (UA, 99%)".

7. Line 116: it should be clarified how GCE was prepared before modification.

8. Lines 128-130: it is necessary to specify the accelerating voltage value for SEM and TEM. The brand and manufacturer for the TEM microscope and EDS detector should also be specified.

9. Figure 1: replace "Electrochemica" with "Electrochemical".

10. Line 170: "(f)" is missing.

11. Lines 193-194: should be "5 mM DA" instead of "5 M DA".

12. Lines 225-240: provide values for all parameters when calculating the number of electrons involved in the electrochemical oxidation of dopamine. Support your conclusions with literary references.

13. Line 238: "quasi-reversible process" is reported. However, the cyclic curves in Figs. 5e, 7a and 7c characterize the electrochemical process as irreversible.

14. Figure 5a: provide DPV-response of bare GCE in the presence/absence of DA.

15. Table 1 is not described in the text.

16. Give the dependence of the dopamine oxidation current on the pH of the solution.

17. It is necessary to present the results of testing the proposed sensor in the analysis of model solutions or real samples with an assessment of the measurement accuracy.

Author Response

Dear Reviewer:

Thank you for reviewing our manuscript “Pd Nanoparticles Loaded on Cu Nanoplates Sensor for Ultrasensitive Detection of Dopamine” (No.: Sensors-3105186) and providing many constructive suggestions which have strengthened our manuscript. We have now carefully revised the manuscript according to the comments.

Comments 1: The Introduction should discuss recent advances in electrochemical detection of dopamine. For example, see the following reviews: Biosensors, 2021, 11(6), 179, https://doi.org/10.3390/bios11060179; Trends in Analytical Chemistry, 2023, 169, 117367, https://doi.org/10.1016/j.trac.2023.117367; Sensors and Diagnostics, 2023, 2, 559-581, https://doi.org/10.1039/D2SD00230B; Journal of Electroanalytical Chemistry, 2024, 959, 118157, https://doi.org/10.1016/j.jelechem.2024.118157; ECS Sensors Plus, 2024, 3(2), 020601, https://doi.org/10.1149/2754-2726/ad3950.

Response 1: We thank the reviewer for the kind advice. In this revised manuscript, we have discussed recent advances in electrochemical detection of dopamine as you suggested.

  1. Karim, A.; Yasser, M.; Ahmad, A.; Natsir, H.; Wahid Wahab, A.; Fauziah, S.; Taba, P.; Pratama, I.; Rosalin; Rajab, A.; Nur Fitriani Abubakar, A.; Widayati Putri, T.; Munadi, R.; Fudhail Majid, A.; Nur, A.; Fadliah; Rifai, A.; Syahrir, M. J. Electroanal. Chem. 2024, 959, 118157.
  2. Burns, G.; Ali, M. Y.; Howlader, M. M. R. TrAC Trends in Anal. Chem. 2023, 169, 117367.
  3. Delmo, N.; Mostafiz, B.; Ross, A. E.; Suni, J.; Peltola, E. Sensors & Diagnostics 2023, 2, (3)559-581.
  4. Lakard, S.; Pavel, I. A.; Lakard, B. Biosensors (Basel) 2021, 11, (6)179.
  5. DeVoe, E.; Andreescu, S. ECS Sensors Plus 2024, 3, (2)020601.

Comments 2: Throughout the entire manuscript, it is necessary to adhere to the unambiguous designation of the glassy carbon electrode (GCE) and the sensor developed on its basis. Note that line 182 uses "Cu@Pd" while Table 1 uses "Pd@Cu/GC".

Response 2: We thank the reviewer for the constructive comment. We have carefully made the appropriate corrections as you suggested.

Before modification:

“Nevertheless, the excessive and aggregated Pd nanoparticles cannot reduce the use amount of noble metal, and may also affect the catalytic performance of Cu@Pd nanoplates even at a higher Pd loading.”

After modification:

“Nevertheless, the excessive and aggregated Pd nanoparticles cannot reduce the use amount of noble metal, and may also affect the catalytic performance of Pd/Cu NPTs even at a higher Pd loading.” (Lines 199)

Comments 3: Lines 46-47: note that Ulubay et al. used "nano-Cu/PPy/GCE" and "human urine samples" instead of "Cu/GC" and "human blood serum".

Response 3: We would like to thank this reviewer’s kind suggestion. In this revised manuscript, "nano-Cu/PPy/GCE" and "human urine samples" have been used instead of "Cu/GC" and "human blood serum". (Lines 46-47)

Before modification:

 “Sükriye Ulubay et al. reported the application of Cu/GC electrodes for the electrochemical detection of DA in the experimental and real samples (human blood serum).”

After modification:

Sükriye Ulubay et al. reported the application of nano-Cu/PPy/GCE for the electrochemical detection of DA in the experimental and real samples (human urine samples).

Comments 4: Line 74: note that Huang et al. used "Pd/CNF-CPE" instead of "Pd/CNFs modified GC".

Response 4: We would like to thank this reviewer’s kind suggestion. In this revised manuscript, " Pd/CNF-CPE " have been used instead of "Pd/CNFs modified GC".

Before modification:

“Huang J. et al prepared carbon nanofibers (CNF) loaded with Pd nanoparticles by electrospinning and heat treatment processes. The constructed Pd/CNFs modified GC has good electrochemical catalytic activity for DA, uric acid (UA), and AA.”

After modification:

“Huang J. et al prepared carbon nanofibers (CNF) loaded with Pd nanoparticles by electrospinning and heat treatment processes. The constructed Pd/CNF-CPE has good electrochemical catalytic activity for DA, uric acid (UA), and AA.” (Line 75)

Comments 5: The abbreviations PVP (line 89), PBS (line 125) and Glu (line 254) should be identified. Subsection 2.1. "Reagents" does not contain information on "Glu".

Response 5: We would like to thank this reviewer’s kind suggestion. In this revised manuscript. The abbreviations PVP (line 89), PBS (line 125) and Glu (line 254) have been identified.

Before modification:

“PVP (MW: 10,000)”

After modification:

“polyvinyl pyrrolidone (PVP, MW: 10000)”, glucose (Glu, C6H12O6, 99%),

Before modification:

PBS solution (0.2 M, pH=7.4)

After modification:

phosphate-buffered saline (PBS, 0.2 M, pH=7.4)

Comments 6: Line 93: "uric acid (C5H4N4O3, 99%)" should be replaced with "uric acid (UA, 99%)".

Response 6: We would like to thank this reviewer’s kind suggestion. In this revised manuscript, "uric acid (C5H4N4O3, 99%)" have been replaced with "uric acid (UA, 99%)".

Before modification:

“hydrogen peroxide(H2O2, 30wt%, wt%), uric acid (C5H4N4O3, 99%), and branched polyethyleneimine (BPEI, 50wt%, MW: 60,000) were obtained from InnoChem and used without purification.”

After modification:

“”hydrogen peroxide(H2O2, 30wt%, wt%), uric acid (UA, 99%), and branched polyethyleneimine (BPEI, 50wt%, MW: 60,000) were obtained from InnoChem and used without purification.”(line 96).

Comments 7: Line 116: it should be clarified how GCE was prepared before modification.

Response 7: We would like to thank this reviewer’s kind suggestion. In this revised manuscript, we have clarified how GCE was prepared before modification.

“The pretreated GCE is prepared as follows: GC electrode was polished with 5.0, 1.0, 0.3 μm Al2O3 slurry in turn and washed ultrasonically with water and ethanol for 5 mins, respectively.” (Line 119-121)

Comments 8: Lines 128-130: it is necessary to specify the accelerating voltage value for SEM and TEM. The brand and manufacturer for the TEM microscope and EDS detector should also be specified.

Response 8: We thank the reviewer for the kind advice. In this revised manuscript, we have carefully made the appropriate corrections as you suggested.

Before modification:

 “The morphology and elemental distribution of Pd/Cu NPTs was observed by scanning electron microscopy (SEM, Zeiss Gemini 300, Oberkochen, Germany) and transmission electron microscope (TEM) with energy-dispersive spectroscopy (TEM-EDS).”

After modification:

“The morphology and elemental distribution of Pd/Cu NPTs was observed by scanning electron microscopy (SEM, Zeiss Gemini 300, at 200 kV, Oberkochen, Germany) and transmission electron microscope (TEM, at 200 kV, JEOL JEM-2100) with energy-dispersive spectroscopy (TEM-EDS, at 200 kV, JEOL JEM-2100).”(line 134-137)

Comments 9: Figure 1: replace "Electrochemica" with "Electrochemical".

Response 9: We thank the reviewer for the kind advice. We've replaced "Electrochemica" with "Electrochemical".

Comments 10: Line 170: "(f)" is missing.

Response 10: We thank the reviewer for the kind advice. We've added "(f)" to it.

Before modification:

Figure 3. (a) TEM image and (b) high resolution TEM image of the Pd/Cu NPTs; TEM-EDS mapping images of Cu (c) and Pd (d), (e) TEM-EDS profile and XRD patterns of Pd/Cu NPTs.

After modification:

Figure 3. (a) TEM image and (b) high resolution TEM image of the Pd/Cu NPTs; TEM-EDS mapping images of Cu (c) and Pd (d), (e) TEM-EDS profile and (f) XRD patterns of Pd/Cu NPTs.

Comments 11: Lines 193-194: should be "5 mM DA" instead of "5 M DA".

Response 11: We thank the reviewer for the constructive comment. In this revised manuscript, "5 mM DA" have been instead of "5 M DA".

Before modification: “As the test is conducted in the presence of DA (5 M)”

After modification: “As the test is conducted in the presence of DA (5 mM)”

Comments 12: Lines 225-240: provide values for all parameters when calculating the number of electrons involved in the electrochemical oxidation of dopamine. Support your conclusions with literary references.

Response 12: We thank the reviewer for the constructive comment. In this revised manuscript, we added literary references to support our conclusions.

Comments 13: Line 238: "quasi-reversible process" is reported. However, the cyclic curves in Figs. 5e, 7a and 7c characterize the electrochemical process as irreversible.

Response 13: We thank the reviewer for the constructive comment. Please excuse our error of expression. In this revised manuscript, we changed "quasi-reversible process" to "irreversible" according to your suggestion.

Comments 14: Figure 5a: provide DPV-response of bare GCE in the presence/absence of DA.

Response 14: We thank the reviewer for the constructive comment. In this revised manuscript, we provide DPV-response of bare GCE in the presence/absence of DA (Fig. S11). In order for readers to better understand the full text, we include the description of this part in the article. (Line 218)

Comments 15: Table 1 is not described in the text.

Response 15: We thank the reviewer for the constructive comment. In this revised manuscript,

we added the description of Table 1.

“Compared with previously reported electrodes (Table 1), the developed Pd/Cu NPTs/GCE shows superior performance with a lower detection limit of 0.1045 μM in the concentration range of 0.047-1.12 mM.” (Lines 273-275)

Comments 16: Give the dependence of the dopamine oxidation current on the pH of the solution.

Response 16: We thank the reviewer for the constructive comment. In this revised manuscript, we added the dependence of the dopamine oxidation current on the pH of the solution (Fig. S10). From the figure, we can see that pH=7.2 is the most appropriate. In order for readers to better understand the full text, we include the description of this part in the article. (Line 208-210)

Comments 17: It is necessary to present the results of testing the proposed sensor in the analysis of model solutions or real samples with an assessment of the measurement accuracy.

Response 17: We would like to thank this reviewer’s kind suggestion. We added the analysis of a real sample for dopamine measurement using Pd/Cu NPTs/GCE (Table S2) . In order for readers to better understand the full text, we include the description of this part in the “3.5. Detection in human serum sample”. (Line 275)

Reviewer 3 Report

Comments and Suggestions for Authors

The paper entitled “Pd nanoparticles loaded on Cu nanoplates sensor for ultrasensitive detection of dopamine” by Tan et al. described the development of an electrochemical sensor for dopamine based on palladium nanoparticles loaded on Cu nanoplate and deposited on the surface of GCE . The results are interesting; however, several questions still can be addressed as follows:

1.          What is the function of using BPEI to form Pd nanoplates?

2.          Based on Figure 2a, is it possible to measure the shifting of maximum wavelength from the solution of Cu nanoplate into Pd/Cu nanoplate solution?

3.          Based on Figure 2b, what is the size of synthesized Cu nanoplates? Is it possible to calculate the size distribution of Cu nanoplates?

4.          According to Figure 2d, what is the small nanoparticles with less than 10 nm? Could the EDS mapping element be displayed from the Cu/Pd nanoplates solution to distinguish between Pd nanoparticles and Cu nanoplates?

5.          There is a typo in line 160, it should be TEM-EDS

6.          Based on Figure 3b of the HR-TEM image obtained from Pd/Cu nanoplates, is it possible to obtain the lattice distance from Cu nanoplates?

7.          Based on the TEM analysis, is it possible to calculate the size distribution of Pd nanoparticles on Cu nanoplates?

8.          Is it possible to perform the analysis of the EDS mapping element from the SEM result as shown in Figure 4?

9.          Based on Figure 5b, is it possible to calculate the electrode resistance of Cu/GCE and Pd/Cu NPTs/GCE from the result of impedance analysis?

10.       Based on Figure 5c&d, is it possible to perform a TEM analysis of Pd/Cu NPTs-20 which was given the highest electrocatalytic activity towards dopamine oxidation?

11.       The authors are suggested to calculate the electrochemically active surface area (ECSA) of Pd/Cu NPTs/GCE based on the result shown in Figure 5e. The calculated ECSA of this electrode can be compared with the Cu/GCE and bare GCE.

12.       Is there any pH effect toward dopamine oxidation when it was investigated using Pd/Cu NPTs/GCE?

13.       Is it possible to calculate the kinetic parameters of the electrochemical oxidation process of dopamine such as 1) electron transfer rate constant (ks) and catalytic rate constant (kh) when it was measured with Pd/Cu NPTs/GCE?

14.       Is it possible to calculate a diffusion coefficient (D) of dopamine by performing the chronoamperometric studies using Pd/Cu NPTs/GCE?

15.       What is the sensitivity of dopamine measurement using Pd/Cu NPTs/GCE?

16.       Based on Figure 6, why is the current of DA oxidation facing downward in the first time of addition of dopamine? What is the concentration of added dopamine after adding interfering substances?

17.       Is it possible to quantify the current changes when dopamine was added with potential interfering substances as shown in Figure 6?

18.       Is it possible to perform the analysis of a real sample for dopamine measurement using Pd/Cu NPTs/GCE?

Comments on the Quality of English Language

English is fine, just minor corrections to the manuscript

Author Response

General Comments: Comments and Suggestions for Authors:

The paper entitled “Pd nanoparticles loaded on Cu nanoplates sensor for ultrasensitive detection of dopamine” by Tan et al. described the development of an electrochemical sensor for dopamine based on palladium nanoparticles loaded on Cu nanoplate and deposited on the surface of GCE . The results are interesting; however, several questions still can be addressed as follows:

Response: We appreciate your positive comments.

Comments 1: What is the function of using BPEI to form Pd nanoplates?

Response 1: We would like to thank this reviewer’s kind suggestion. Branched polyethyleneimine (BPEI, 50wt%, MW: 60,000) was used as a kind of stabilizer to form Pd nanoplates. The BPEI was utilized to synthesize Cu nanoplates due to it contains a large number of amine group. Amine group was included in BPEI and acted as a stabilizer by forming s stable complex with Cu(II) ion.

The report of BPEI as a stabilizer can be referred to:

RSC Adv., 2020,10, 44601-44610.

Chemical Physics Letters, 2013, 575, 71-75.

Chemical Physics Letters, 2014, 592, 265-271.

Comments 2: Based on Figure 2a, is it possible to measure the shifting of maximum wavelength from the solution of Cu nanoplate into Pd/Cu nanoplate solution?

Response 2: We would like to thank this reviewer’s kind suggestion. According to your suggestions, we have made corresponding tests and obtained relevant UV-vis absorption spectra of Cu nanoplates and Pd/Cu NPTs suspensions (shown in Fig. S2). Commonly, the Cu nanoparticles suspension shows an absorption peak at around 600 nm (ACS Appl. Mater. Interfaces 2013, 5, 3839−3846). However, UV−vis absorption spectra of the Cu nanoplates suspension (the black curve in Fig. S2) and Pd/Cu NPTs suspension (the red curve in Fig. S2) did not present any UV absorption peak due to too large size of Cu. Therefore, there is no corresponding Cu absorption peak in either material.

Comments 3: Based on Figure 2b, what is the size of synthesized Cu nanoplates? Is it possible to calculate the size distribution of Cu nanoplates?

Response 3: We would like to thank this reviewer’s kind suggestion. We did SEM characterization again according to your suggestion. The Cu nanoplates was measured in the low resolution condition. The length size of sides from Cu nanoplates is in the range of 3-27 μm (Fig. S3a). The thickness of Cu nanoplates around 270 nm (Fig. S3b). The corresponding Cu size distribution is obtained by calculating 200 Cu nanoplates (Fig. S4). The size of synthesized Cu nanoplates is around 14.6±1.0 μm.

Comments 4: According to Figure 2d, what is the small nanoparticles with less than 10 nm? Could the EDS mapping element be displayed from the Cu/Pd nanoplates solution to distinguish between Pd nanoparticles and Cu nanoplates?

Response 4: We would like to thank this reviewer’s kind suggestion.

Comments 4-1. According to Figure 2d, what is the small nanoparticles with less than 10 nm?

Response 4-1: The small nanoparticles with less than 10 nm are Pd nanoparticles and a small amount of Cu nanosheets left after etching.

Comments 4-2. Could the EDS mapping element be displayed from the Cu/Pd nanoplates solution to distinguish between Pd nanoparticles and Cu nanoplates?

Response 4-2: According to your suggestion, we re-characterized TEM+EDS mapping elements, and Pd nanoparticles and Cu nanosheets can be clearly distinguished from the Fig. S5.

Comments 5: There is a typo in line 160, it should be TEM-EDS.

Response 5: We would like to thank this reviewer’s kind suggestion. In this revised manuscript, we have carefully made the appropriate corrections as you suggested.

Before modification: “The TME-EDS analysis was employed to characterize the distribution of red spots with a high density in Fig. 3c indicates the presence of Cu and the scattered distribution of the green spots in Fig. 3d indicates the presence of a relatively low concentration of Pd loading.”

After modification: “The TEM-EDS analysis was employed to characterize the distribution of red spots with a high density in Fig. 3c indicates the presence of Cu and the scattered distribution of the green spots in Fig. 3d indicates the presence of a relatively low concentration of Pd loading.”

Comments 6: Based on Figure 3b of the HR-TEM image obtained from Pd/Cu nanoplates, is it possible to obtain the lattice distance from Cu nanoplates?

Response 6: We would like to thank this reviewer’s kind suggestion. We re-performed the HR-TEM characterization to obtain the lattice distance from Cu nanoplates (Fig. S7). In order for readers to better understand the full text, we include the description of this part in the article. (Line 178-179)

Comments 7:. Based on the TEM analysis, is it possible to calculate the size distribution of Pd nanoparticles on Cu nanoplates?

Response 7: We would like to thank this reviewer’s kind suggestion. According to your suggestion, the size distribution of Pd nanoparticles on Cu nanoplates was calculated and the average size is about 3.6 nm (Fig. 3d). In order for readers to better understand the full text, we include the description of this part in the article. (Line 177-178).

Comments 8:. Is it possible to perform the analysis of the EDS mapping element from the SEM result as shown in Figure 4?

Response 8: We would like to thank this reviewer’s kind suggestion. According to your suggestion, we re-characterized TEM images (Fig. S8) and STEM+EDS mapping elements of Pd/Cu NPTs, and Pd nanoparticles and Cu nanosheets can be clearly distinguished from the Fig. S9. We can see from the figure that most of the fine particles are Pd nanoparticles, and a small part of them are Cu nanosheets.

Comments 9: Based on Figure 5b, is it possible to calculate the electrode resistance of Cu/GCE and Pd/Cu NPTs/GCE from the result of impedance analysis?

Response 9: We would like to thank this reviewer’s kind suggestion. In this revised manuscript, we have calculated the electrode resistance of Cu/GCE and Pd/Cu NPTs/GCE from the result of impedance analysis. The electrode resistances of Cu/GCE and Pd/Cu NPTs/GCE is 49.6 and 48.6 Ω respectively. We include the corresponding conclusions in the text.

“Fig. 5b show that the Pd/Cu NPTs/GCE and Cu/GCE electrodes have litter difference in internal resistance. The electrode resistances of Cu/GCE and Pd/Cu NPTs/GCE is calculated to 49.6 and 48.6 Ω, respectively.” (Line 218-220)

Comments 10: Based on Figure 5c&d, is it possible to perform a TEM analysis of Pd/Cu NPTs-20 which was given the highest electrocatalytic activity towards dopamine oxidation?

Response 10: We would like to thank this reviewer’s kind suggestion. According to your suggestion, we re-characterized TEM images of Pd/Cu NPTs-20 (Fig. S12). The higher catalytic activity is attributed to the loading of more Pd nanoparticles. In order for readers to better understand the full text, we include the description of this part in the article. (Line 238-239).

Comments 11: The authors are suggested to calculate the electrochemically active surface area (ECSA) of Pd/Cu NPTs/GCE based on the result shown in Figure 5e. The calculated ECSA of this electrode can be compared with the Cu/GCE and bare GCE.

Response 11: We thank the reviewer for the constructive comment. The electrochemically active surface area (ECSA) is calculated using the following formula:

ipa= 2.69×105n3/2D1/2Aυ 1/2

In the equation, ipa= the peak current intensity (A), n= the number of participating electrons (n = 2), D = diffusion coefficients of the probe molecule (D = 1.30 × 10−6 ), υ = scan rate (50 V⋅s−1 )

A= electrochemically active surface area (ECSA). So, electrochemically active surface area (ECSA) was measured with bare GCE, Cu/GCE and Pd/Cu NPTs/GCE are 0.0702, 0.071 and 0.0912 cm2 , respectively (Fig. S13). The above calculated data is obtained by estimation, we apologize if there is any deviation.

Comments 12: Is there any pH effect toward dopamine oxidation when it was investigated using Pd/Cu NPTs/GCE?

Response 12: We thank the reviewer for the constructive comment. In this revised manuscript, we added the dependence of the dopamine oxidation current on the pH of the solution (Fig. S10). From the figure, we can see that pH=7.2 is the most appropriate. In order for readers to better understand the full text, we include the description of this part in the supporting information. (Line 208-210)

Comments 13: Is it possible to calculate the kinetic parameters of the electrochemical oxidation process of dopamine such as 1) electron transfer rate constant (ks) and catalytic rate constant (kh) when it was measured with Pd/Cu NPTs/GCE?

Response 13: We would like to thank this reviewer’s kind suggestion.

1: According to Laviron theory, the electron transfer rate constant (ks) reaches 10.62 s-1.

Laviron equation: Epc=E0+RTln(RTks/αnF)/αnFRTlnv/αnF

Epc is the equilibrium potential (0.25V), E0 is the standard electrode potential (0.1 V), R is the ‌gas constant ( 8.314 J mol-1·K-1), T is the absolute temperature (298.15 K), ks is the reaction rate constant, α is the number of electron transfers (2), n is the number of electrons involved in the reaction (2), F is ‌Faraday's constant, V is ‌scan rate (50 mV/s).

2: Catalytic rate constant (kh) also known as turnover number:

Formula: TON= Reaction electron number/Active site number,

So, the catalytic rate constant (kh) was measured with Pd/Cu NPTs/GCE is 28.6 mmol·h⁻¹·g⁻¹.

The above calculated data is obtained by estimation, we apologize if there is any deviation.

Comments 14: Is it possible to calculate a diffusion coefficient (D) of dopamine by performing the chronoamperometric studies using Pd/Cu NPTs/GCE?

Response 14: We would like to thank this reviewer’s kind suggestion.

Diffusion coefficient (D) formula:

D=(kT/q)μ

Where k is the Boltzmann constant (k=1.380649 ×10-23 J/K), T is the absolute temperature (T=298.15 K), q is the amount of charge, and μ is the mobility.

So, the Diffusion coefficient (D) was measured with Pd/Cu NPTs/GCE is 1.30 × 10−6 cm2 ⋅s−1.

The above calculated data is obtained by estimation, we apologize if there is any deviation.

Comments 15: What is the sensitivity of dopamine measurement using Pd/Cu NPTs/GCE?

Response 15: We would like to thank this reviewer’s kind suggestion. We recalculated the sensitivity as you suggested. The sensitivity of Pd/Cu NPTs/GC electrode to dopamine was calculated to be 218.43 μA· mM-1·cm-2. In order for readers to better understand the full text, we have made some changes in the presentation of the article.

“The LOD is calculated to be 0.1045 μM.” changed to “The sensitivity and LOD of Pd/Cu NPTs/GC electrode to dopamine were calculated to be 218.43 μA· mM-1·cm-2 and 0.1045 μM, respectively.” (Line 272-275)

Comments 16: Based on Figure 6, why is the current of DA oxidation facing downward in the first time of addition of dopamine? What is the concentration of added dopamine after adding interfering substances?

Response 16: We would like to thank this reviewer’s kind suggestion.

Comments 16-1: What is the concentration of added dopamine after adding interfering substances?

Response 16-1: We added 1.0 mM dopamine (10 μL) after adding interfering substances.

Comments 16-2: Based on Figure 6, why is the current of DA oxidation facing downward in the first time of addition of dopamine?

Response 16-2: The reason for the current decline may be that the amount of DA added is relatively small, (10 uL DA added to 10 mL electrolyte solution, the concentration of DA obtained is relatively small), in addition, other interfering substances have been added to the solution, and eventually lead to the current decline

Comments 17: Is it possible to quantify the current changes when dopamine was added with potential interfering substances as shown in Figure 6?

Response 17: We would like to thank this reviewer’s kind suggestion. We included in Table S2 the changes produced when potential interfering substances were added to dopamine.

It can be seen from the figure that when GLU, NaCl and UA are added, the current decreases by 1.3, 0.9 and 1.2 μA, respectively while the addition of AA increases the current by 2.2 μA. This suggests that the addition of the interfering substances slightly affected the sensor's detection of dopamine. In order for readers to better understand the full text, we include the Table S2 in the Supporting information.

Comments 18: Is it possible to perform the analysis of a real sample for dopamine measurement using Pd/Cu NPTs/GCE?

Response 18: We would like to thank this reviewer’s kind suggestion.we added the analysis of a real sample for dopamine measurement using Pd/Cu NPTs/GCE (Table S1) . In order for readers to better understand the full text, we include the description of this part in the “3.5. Detection in human serum sample”. (Line 275)

Round 2

Reviewer 1 Report

Comments and Suggestions for Authors

Even if the authors claimed the improvement of this synthetic process of Cu/Pt composite, it does not make sense for the enhanced catalytic property. There is no dicussion for the reason at all, the whole paper was demonstrating the approach, not providing detailed reasons or enough supporting proofs, therefore can not provide sufficient support for the conclusions claimed.

Besides, for the innovation, the reviewer feeeeels very difficult to find any....

Comments on the Quality of English Language

Could be improved.

Author Response

Comment: Even if the authors claimed the improvement of this synthetic process of Cu/Pt composite, it does not make sense for the enhanced catalytic property. There is no dicussion for the reason at all, the whole paper was demonstrating the approach, not providing detailed reasons or enough supporting proofs, therefore can not provide sufficient support for the conclusions claimed. Besides, for the innovation, the reviewer feeeeels very difficult to find any....

Answer: Thanks for these comments. According to your suggestion, the relevant mechanism has been added in the discussion on improving the detection efficiency of Pd/Cu NPTs. The preparation of composite materials to achieve the synergistic catalytic performance of two catalytic materials is a verified effective method. The synergistic catalysis performances, arising from the combined effects of catalytic superposition, interface interactions, surface electron effect, also are affected by the composition and structure of composite materials. Despite extensive research on Cu/Pd alloy or composite materials for catalysis, the catalytic properties of composites with different structures remain unpredictable, and the synthesis process significantly influences the structure of these materials. This paper optimized a new approach for preparing island-structured Pd/Cu NPTs with excellent catalytic performance. The significance of this research resides in validating the enhanced stability and catalytic activity of the catalyst (Pd/Cu NPTs). Therefore, we agree with the reviewer’s evaluation that there is insufficient innovation. However, we still insist that this work has certain research significance.

In our work, loading a small amount of Pd metal on their surface firstly is intended to safeguard the relatively unstable Cu inside with the relatively stable Pd. Secondly, while preserving the catalytic activity of copper nanoplates, it can additionally increase the catalytic active center with Pd, thus attaining an enhanced catalytic activity of the Pd/Cu NPTs for electro-catalytic oxidation of dopamine. This is entirely different from alloys. Our results also confirmed that Pd/Cu NPTs has a better catalytic activity and stability compared to Cu nanoplates. In the future, we will continue to explore mechanisms, such as theoretical calculation to investigate the influence of interface effects on the catalytic properties of island-structured Pd/Cu NPTs. The method proposed in this paper will also be applied to a wider range of composite catalytic materials for further exploration of the structure-activity relationship between microscopic morphology and catalytic performance, aiming to yield more innovative research. Eventually, by the way, we made Pd/Cu NPTs rather than Cu/Pt composites.

For detailed modification, please refer to the revised manuscript. (Please refer to line 210 on page 7, line 213-214 on page 7, line 231-232 on page 8, and 293-295 on page 9). Please see the following list.

(1) “Cu/GCE electrode shows a new characteristic oxidation peak to DA at a potential of around 0.2 V, indicating Cu/GCE electrode can recognize the presence of DA” has been revised to “Cu/GCE electrode shows a new characteristic oxidation peak to DA at a potential of around 0.2 V, indicating Cu nanoplates have a catalytic effect on the oxidation of dopamine and Cu/GCE electrode also can recognize the presence of DA.” (Please refer to line 210 on page 7)

(2) “Obviously, the addition of Pd improves the response performance of the sensor to DA.” has been revised to “Obviously, the addition of Pd improves the response performance of the sensor to DA due to the newly added active sites centered on Pd on the surface of Cu nanoplates.” (Please refer to line 213-214 on page 7)

(3) The sentence of “Compared with the aggregation of Pd in Pd/Cu NPTs-70, Pd/Cu NPTs-20  may be capable of providing more effective active sites” has been added in this revised manuscript .( please refer to line 231-232 on page 8)

(4) The sentence of “It may be attributed to the relatively stable Pd loaded on the surface of Cu nanoplates, eventually improving the stability of the Pd/Cu NPTs in the electrochemical sensing DA” has been added in this revised manuscript (please refer to 293-295 on page 9)

Reviewer 2 Report

Comments and Suggestions for Authors

The authors responded and worked through all the questions and comments, significantly improving the manuscript. I recommend the article for publication in its current version.

Author Response

Comment:The authors responded and worked through all the questions and comments, significantly improving the manuscript. I recommend the article for publication in its current version.

Answer: we are grateful for the reviewer’s affirmation and the constructive suggestion in the last review.